# The Effects of Oral Lavender Therapy on Wounding in Chimpanzees (*Pan troglodytes*)

**DOI:** 10.3390/ani13081382

**Published:** 2023-04-18

**Authors:** Elizabeth R. Magden, Sarah Neal Webb, Susan P. Lambeth, Stephanie J. Buchl, Steven J. Schapiro

**Affiliations:** Michale E. Keeling Center for Comparative Medicine and Research, The University of Texas MD Anderson Cancer Center, 650 Cool Water Dr., Bastrop, TX 78602, USA

**Keywords:** lavender, nonhuman primates, chimpanzees, conflict, wounding

## Abstract

**Simple Summary:**

Lavender (*Lavandula angustifolia*) used as a therapeutic has been shown to alleviate anxiety and stress in humans but has yet to be studied in nonhuman primates (NHPs). In this study, we administered lavender in the form of oral capsules as a possible therapy to decrease wounding in a subset of chimpanzees, with the thought that decreasing anxiety and stress may also decrease conflict and, therefore, wounding. Overall, wounding did not decrease with lavender therapy; however, the percentage of wounds requiring medical intervention did decrease with lavender therapy.

**Abstract:**

Lavender administration in humans has been shown to promote calmness without the side effects often observed with benzodiazepines. Studies in both humans and rodents have found that ingestion of oral lavender capsules resulted in significantly decreased anxiety. Additionally, mice developed an anti-conflict effect and humans increased socially inclusive behaviors. Given the safety of oral lavender oil and the observed benefits, we administered daily lavender capsules to six chimpanzees who exhibited conflict-instigating behaviors in an effort to further decrease our already low levels of wounding. We compared the total number of wounds in 25 chimpanzees housed with the six lavender-treated chimpanzees in five different social groups (1) prior to administration of daily oral lavender capsules to (2) total wounds during daily oral lavender capsule treatment. We hypothesized that lavender therapy treatment would reduce overall wounding in the social groups. Surprisingly, overall wounding was higher during the lavender treatment period (*p* = 0.01), yet the percentage of wounds requiring treatment significantly decreased during the lavender therapy period (36% vs. 21%, *p* = 0.02).

## 1. Introduction

Lavender is one of the most common phytotherapeutic remedies for anxiety and stress [1]. There are four species of this Mediterranean evergreen perennial plant, but the one most commonly used for therapeutic purposes is *Lavandula angustifolia* [2]. This particular species has been found to be more effective at reducing stress in comparison to other lavender species, such as *L. stoechas* [3]. Lavender has a long history of medicinal use, cited often in Roman literature, such as “De Materia Medica” by Dioscorides from the 16th century [2]. In the Middle Ages, lavender was coveted as a valuable essential oil, used to make perfume and soap [4]. In fact, the origin of the word lavender arises from the Latin verb “lavare”, which means “to wash”. Lavender was also used as an antimicrobial agent during the First World War [5]. The scent of lavender has been associated with positive emotional states in addition to the observed therapeutic benefits. The benefits are believed to arise from both the psychological effect of the odor as well as the physiological effect of the inhaled volatile compounds found in lavender oil [6]. The essential oil has gained popularity in both cosmetic and complementary medicine applications based on its numerous reported benefits, low cost, and high safety index.

The essential oils are gathered from the lavender flowers, which are collected prior to blooming. The oils are obtained via distillation from the surface of the flower calyx, and the most abundant chemical compounds in these distilled oils are linalool and linalyl acetate [2]. Since lavender can lose over 40% of its essential oils with drying, a fluidized bed can be utilized that is composed of a closed-circuit system with a desiccant and heat exchanger in order to preserve the biological activity of the lavender [4,7]. Lavender grows best in a hot and dry climate [5]. Harvest time can also affect the phenolic content of lavender, and plants harvested in summer and autumn have shown increased levels of essential oil compared to those plants harvested in winter and spring [8]. Lavender is edible, and lavender tinctures are used to treat a variety of ailments, as it has both antioxidant and antimicrobial properties [2]. It can be used as an antiseptic and has antifungal activity [9,10]. It has also been used extensively to alleviate stress and anxiety [2,11,12,13,14,15,16,17]. Anxiety disorders affect approximately 10–30% of the adult human population, with prevalence higher in females compared to males [1,18]. Medications typically used to help alleviate anxiety, such as benzodiazepines, have potential side effects that include habituation, abuse, sedation, and withdrawal symptoms [19]. Lavender oil appears to have a calming effect without producing sedation, dependence, tolerance, or withdrawal, which gives it an advantage over benzodiazepines [19]. Indeed, when compared to benzodiazepines, one study found no significant difference between lavender and benzodiazepine treatments for self-rated measures of anxiety [16].

Lavender oil can be used as a topical oil, inhaled as aromatherapy, or ingested as an oral capsule. Studies examining inhaled lavender as aromatherapy found that it increased parasympathetic tone and relaxation and increased arousal levels in human subjects [20]. Atsumi and Tonosaki (2007) reported that sniffing lavender for 5 min increased free-radical scavenging activity, which protected cells from damage, and decreased salivary cortisol levels, which decreased overall stress and protected the body from oxidative stress [21]. Lavender inhalation has also been shown to reduce salivary chromogranin A (a marker of stress) following exposure to a stress-inducing arithmetic task [22].

Oral lavender is available as an over-the-counter dietary supplement (e.g., CalmAid^®^, Nature’s Way, Green Bay, WI, USA with a recommended dose of an 80 mg capsule for administration once or twice daily. CalmAid^®^ contains linalool (36.8%) and linalyl acetate (34.2%), two of the primary constituents of lavender oil [19]. Various studies using oral lavender oil capsules (CalmAid^®^) for at least 6 weeks found significant decreases in anxiety in human patients [23,24,25,26,27].

In rodents, rats exposed to lavender oil have shown decreased anxiety, as measured by the elevated plus maze, forced swimming tests, and Geller and Vogel tests [28,29]. In mice exposed to linalool, Linck and colleagues (2010) observed decreased anxiety, increased social interaction, and decreased aggressive behavior. In other rodent studies, lavender oil had similar anti-anxiety effects when compared to chlordiazepoxide and diazepam [30,31]. Two studies reported that lavender even had an anti-conflict effect in mice [29,32], an effect that is also found in humans. Research shows lavender and other calming scents promote inclusive behaviors compared to stimulating scents, like peppermint, which promote exclusive states [33]. In one study, participants played the Trust Game, where the first player can choose an amount of money to transfer to an anonymous second player. Those first players exposed to lavender transferred more money to the trustee in comparison to those exposed to peppermint. The results suggest that lavender may induce a higher level of social trust [33]. Lavender has also been shown to attenuate fatigue and to have a positive effect on affection in humans [34,35].

Several studies investigating the mechanism of action for lavender have found that it exerts an affinity for the glutamate N-methyl-D-aspartate (NMDA)-receptor in a dose-dependent manner and also binds to the serotonin transporter (SERT). Therefore, the observed anxiolytic and anti-depressive effects of lavender may be due to antagonism of the NMDA-receptor and inhibition of SERT [1]. Lavender does appear to require chronic administration in order to achieve the maximum anxiolytic benefits and the effects are dose-dependent [16]. This strengthens the argument that observed benefits are pharmacological, given that the more one receives, the stronger the effect [16]. 

Lavender essential oil is generally recognized as “Safe” by the US Food and Drug Administration, meaning it is safe when used as a food additive [36]. While many essential oils cannot be safely administered orally (undiluted) due to inflammatory or cytotoxic effects, lavender oil is well tolerated and can be given undiluted [19]. There is one report of gynecomastia in young boys following topical application, and it is believed lavender may have some estrogenic activity. This side effect was reversed with treatment cessation [31].

Despite the wide safety margin of lavender oil, and the observed benefits found in humans and rodents, we found no studies examining the effects of lavender oil in nonhuman primates (NHPs). Chimpanzees, in particular, can instigate conflict and due to their size and strength this conflict can, at times, result in wounding. Given the natural dominance hierarchy of chimpanzees, there are occasional social altercations as they vie for dominance. These normal social interactions can create social tension, stress, and potential wounding. The behavioral management of captive chimpanzees seeks to enhance their well-being by minimizing such conflict and the potential for wounding [37]. With appropriate care and management, serious conflict is relatively rare in our population of captive chimpanzees, and wounding is even more rare. However, despite the rarity of wounds requiring treatment, any wounding represents a risk to the animals, and we continually search for means to minimize the occurrence in our captive populations. Lambeth et al. (1997) found that the increased presence of personnel on weekdays correlated to increased wounding in chimpanzees, and thus the human effects are now considered part of the colony management strategy [38]. 

Lavender therapy was investigated here as yet another possible tool for minimizing our already low levels of wounding by taking advantage of the anxiolytic and anti-conflict effects of lavender. At the Chimpanzee Care Center, all animals are socially housed, and our observations indicated that in a few of our groups, a single individual appeared to be consistently creating social tension. Therefore, we targeted these conflict instigators as the recipients of lavender therapy. We hypothesized that since lavender has been shown to alleviate stress and anxiety in people and in rodents, it may produce the same effects in the animals that were instigating the social tension. In other words, if we decreased anxiety and stress in the conflict instigators, then social altercations resulting in wounding would also decrease while the target chimpanzees were receiving lavender treatment.

## 2. Materials and Methods

### 2.1. Animals

Six chimpanzees (five males and one female) were selected to receive lavender therapy, as they had been witnessed as instigators of social group tension. Two of these animals (one male—SI, one female—TU) resided in the same social group comprised of six animals. The other four males (NI, AJ, AK, and TO) were in separate, established social groups, ranging in size from four to six chimpanzees. The total number of wounds and percentage of wounds requiring medical treatment were documented for all members (*n* = 25) of each social group. (See Table 1 for demographic information for the treatment subjects and their social partners). 

The chimpanzees at The University of Texas MD Anderson Cancer Center, Michale E. Keeling Center for Comparative Medicine and Research, Chimpanzee Care Center, are maintained in accordance with the *Guide for the Care and Use of Laboratory Animals* [39], and the facilities and program are fully accredited by AAALAC International (Frederick, MD, USA). Animals are housed in mixed-sex, compatible social groups (ranging from 4–10 animals) in large outdoor enclosures with indoor temperature-controlled access. They have ad libitum access to water, receive fresh produce 2–3 times daily, and are fed a high-fiber commercial primate diet (Envigo Teklad #7195, Madison, WI, USA). All chimpanzees participate in an extensive behavioral management program that includes comprehensive environmental enrichment and positive reinforcement training [40,41]. 

### 2.2. Lavender Administration

Lavender therapy consisted of once daily administration of the dietary supplement CalmAid^®^ (Nature’s Way, Green Bay, WI, USA), in which one softgel capsule contains 80 mg of Silexan^™^ English lavender (*Lavandula angustifolia*) essential oil (Figure 1). The capsules were mixed with diluted fruit juice, in which they generally dissolved, and were administered orally to the chimpanzees who are trained to readily accept oral medications. 

### 2.3. Data Aggregation

To examine any differences in wounding as a function of lavender treatment, we created ‘pre-treatment’ periods to match the durations of the ‘during-lavender-treatment’ periods. Each of the six treatment animals received oral lavender for a certain number of days, ranging between 67 and 642 days. The number of pre-treatment days were selected to equal the same number of days as the during-lavender-treatment period. For example, NI and his group had 613 days of pre-treatment and 613 lavender treatment days included in the dataset (Table 1). Wounding observations were standardized by having a small subset of trained veterinary technicians document observed wounding and severity to maintain consistency. Wounding data were aggregated for each chimpanzee receiving lavender oil and for the other chimpanzees within his or her group.

Any trauma or injuries sustained were included in the dataset, including very minor injuries such as abrasions and scratches, and any treatment administered (e.g., antibiotics, topical Vetericyn^®^ hydrogel (Innovacyn, Inc., Rialto, CA, USA), topical honey or sugar paste, hydrotherapy, or laser therapy) for each wound. We counted multiple wounds with the same date (i.e., different types of wounds on different body parts that were likely the result of the same agonistic encounter) separately. Using these criteria, seven pairs of variables were created (i.e., pre-treatment and during treatment for each variable): (1) total number of wounds; (2) number of wounds requiring any treatment; (3) the percentage of wounds requiring any treatment out of the total number of wounds; (4) number of ongoing days (i.e., the number of days that the clinical case continued following wound onset); (5) number of severe wounds (defined as any wound more severe than an abrasion, scratch or swelling); (6) number of wounds requiring antibiotic treatment; and (7) percentage of wounds requiring antibiotic treatment (number of wounds requiring antibiotic treatment/total number of wounds).

### 2.4. Data Analysis

Due to small sample sizes and non-normal data distributions, we used bootstrapped paired-samples *t*-tests to examine differences across the pre-treatment and during lavender treatment periods in the seven pairs of variables described above. We report means (M) and standard errors of the mean (SE) for each variable in Table 2. Analyses were conducted in SPSS 26 (IBM^®^, 2020, Armonk, NY, USA) with *p*-values of 0.05 considered significant.

## 3. Results

Contrary to our initial hypothesis, the total number of wounds during the pre-treatment period was significantly lower than during the treatment period (Table 2; Figure 2a). However, consistent with our expectations, the percentage of wounds requiring treatment (e.g., medication, laser therapy, sugar paste, etc.) during the pre-treatment period was significantly higher than during the lavender-treatment period (Table 2; Figure 2b). There were no other significant differences between pre-treatment and treatment periods (*p* > 0.05).

## 4. Discussion

Lavender therapy was initiated to decrease the already low rate of wounding in our chimpanzee colony. Overall, we observed that the number of wounds increased in chimpanzee social group members during the treatment phase compared to number of wounds during the pre-treatment phase (i.e., prior to the initiation of oral lavender therapy). We do not believe these results necessarily indicate that lavender exacerbates wounding activity, as (1) the evidence to the contrary is abundant and well published and (2) our results also showed that the percentage of wounds requiring treatment decreased following the initiation of lavender therapy. The decrease in the percentage of wounds requiring treatment could indicate that there may be some calming effects of oral lavender therapy.

We focused lavender treatment on a subset of animals, and our animal selection was based on our intimate knowledge of the behavior and social dynamics of the chimpanzees possessed by our animal caregivers, veterinary technicians, behavioral staff, and veterinarians. As this study originated as a presumed therapeutic treatment, we did not include a placebo group, which would have strengthened our results and provided useful comparison data. It would also have been interesting to administer lavender to all colony animals and determine whether overall wounding decreases if the number of animals treated is expanded. Perhaps we did not select the appropriate animals in our initial subset to observe the maximum calming benefits of lavender. We also could have unintentionally affected the delicate social relationships in the chimpanzee social groups by only providing the lavender treatment to the conflict instigators, who generally were the more dominant group members. While we can usually avoid conflict by focusing attention on the dominant animals, it is difficult to predict the subtle social effects of our focused treatments and we cannot rule out that the social group dynamics may have been disrupted by the therapy itself, possibly leading to increased wounding.

In a healthy chimpanzee social colony, there are many factors that can contribute to stress, anxiety, and conflict. Social dynamics frequently shift and can be affected by many outside factors. Changes in the care staff can affect chimpanzee behavior, whether that means a new person in their home area or just a change to the care staff member who provides consistent daily care to the social group. We know that having an increased number of care staff present during the week has been correlated to increased wounding [38]. Other changes outside of our control, or perhaps even our awareness, could have contributed to the increased wounding. This is especially true since much of the data was collected for the groups around the same time frame. These changes could be as simple as the type of produce offered (seasonal availability), the weather (extreme heat vs. cold), or the frequency of outside visitors.

There was not a significant difference in the total number of severe wounds or the percentage of wounds requiring antibiotic treatment. This is likely due to our finding that the overall wounding did not decrease and thus the continued presence of wounds, combined with the low frequency of wounding and our limited sample size, made it difficult to determine any significant differences in these wounding variable categories. It would be beneficial to have a larger sample size, as our lavender recipients consisted of only six animals. Though only six animals received the therapy, the effects were studied in multiple groups (5) that included 19 other animals. Despite our effort to have a robust data set, a larger number of lavender recipients may have been beneficial for finding trends that were not observed in our animal subset.

An additional consideration is that we know that the effects of lavender are dose-dependent and require chronic administration. One study by Malcolm et al. (2017) found that humans who were administered oral lavender at a dose of 160 mg/day showed brain changes when imaged with positron emission tomography and MRI after 8 weeks of therapy, with specifically reduced binding potential at the 5HT1A receptor in the hippocampus and the anterior cingulate cortex [19]. Another study found a significant decrease in anxiety levels in people when dosed at 80 mg/day for at least six weeks [17]. These studies show that it likely takes weeks for oral lavender therapy to have maximum benefit. Since the shortest treatment group in this study received lavender for 67 days, we most likely treated the animals in this study for a long enough duration to observe the initial expected benefits of lavender but possibly not long enough to observe the maximum benefits. We may have needed to increase the dose to twice daily administration and given lavender for longer to appreciate the benefits observed in other studies. While some studies used the 80 mg/day dose of oral lavender and observed a significant decrease in anxiety [17,31], other studies used the higher dose of 160 mg/day in comparison to our dose of 80 mg/day [42]. One meta-analysis reports that those studies using the increased dose of 160 mg/day correlated with a further decrease in anxiety in comparison to studies in people using the lower dose of 80 mg/day, as measured by the Hamilton Anxiety Scale (HAMA) [42]. In our study, the lower dose of 80 mg/day was chosen because it was within the dose range of 80–160 mg/day. Additionally, the oily capsule was not a highly desired item for consumption by the chimpanzees. The oral capsule was dissolved in juice and occasionally was spit out by chimpanzees and had to be re-administered. Our belief is that twice daily administration of the lavender capsule would have led to increased non-compliance with the prescribed supplement. By selecting the dose of one capsule per day we most likely improved compliance with taking the therapy but also perhaps limited some of the beneficial anti-anxiety effects observed in other studies using the higher dose. 

Lavender therapy can also be administered as a topical oil or inhaled in an aerosolized form. A recent meta-analysis done by Ghavami et al. (2022) found that when the data from 21 lavender articles were analyzed, the largest standardized mean difference was achieved with the aromatherapy method when compared to other methods of administration. When inhaled, lavender stimulates the olfactory bulb of the brain and transmits signals to the limbic system, a sensory center that secretes enkephalin, endorphins, and serotonin. It is believed that lavender aromatherapy may reduce stress by reducing the activity of the sympathetic nervous system and increasing activity in the parasympathetic nervous system, which creates a sense of relaxation [3,20,43]. The aromatherapy administration method has also been shown to have almost immediate benefits, with significant reductions in anxiety following the first treatment [4]. Since aromatherapy may work quicker than oral administration, and the aromatherapy method exerts its effects via a different pathway than oral administration, it would be interesting to explore the calming effects of the aromatherapy administration method for lavender in chimpanzees.

Topical lavender oil administration is another therapeutic option. When applied, it absorbs quickly into the skin and the lavender components of linalyl acetate and linalyl glucoside are detectable in the blood five minutes after topical application. Levels peak at 19 min and then decline 90 min after administration [44]. Topical administration has the added benefit of providing antibacterial and antifungal effects when administered at a dose of 4–9 mg/mL [45].

The methods of aromatherapy or topical application would be more challenging to administer to chimpanzees than the oral capsule utilized in this study, but it is possible. All the animals at the Chimpanzee Care Center are trained to voluntarily allow topical medication administration and would voluntarily participate in topical lavender oil applications. Additionally, several animals have been trained to voluntarily accept nebulization therapy (and other types of therapy that require training the chimpanzees to ‘station’) for previous clinical situations [46,47,48], and others could be trained to inhale lavender aromatherapy via nebulization [49].

It should be noted that lavender has additional benefits besides reducing anxiety and could be considered a treatment for these conditions in nonhuman primates, as they have had success in human studies. Lavender has been shown to improve wound healing [50], decrease depression [51], decrease pain associated with labor [52,53,54], help treat local dental infections and prevent the formation of biofilm on teeth [55], lower cholesterol [56,57], regulate blood sugar levels [58], and it may even have anti-cancer effects as it has been shown to inhibit tumor growth by stimulating apoptosis of tumor cells [59,60]. In one study, lavender therapy showed efficacy in treating brain cancer when combined with silver nanoparticles [61]. Lavender has also been shown to have neuroprotective effects, likely due to its antioxidant properties, and may help with Alzheimer’s disease treatment [4,62]. Lavender has even been found to have analgesic and anti-inflammatory properties. Several studies have reported that lavender can decrease post-operative pain [5,63,64], and topical lavender application reduced the pain associated with repeated insertion of needles for patients undergoing hemodialysis [65]. Its analgesic and anti-inflammatory properties, combined with the wide safety margin of lavender, lack of adverse effects, and low cost, make lavender an appealing treatment option for a variety of ailments and conditions [66].

## 5. Conclusions

While oral lavender did not decrease overall wounding, it did decrease the percentage of wounds requiring treatment. It is important to empirically examine whether the therapies we provide are having the desired effects for several reasons: cost, efficiency of staff time, and most importantly, that we do not want to ask the animals to take medications or supplements that are not beneficial. Given the results we observed following oral lavender administration, additional research would be useful in chimpanzees given the large amount of literature supporting lavender as an anxiolytic therapy. Additional studies with larger sample sizes, increased dosing, and other methods of administration (e.g., inhalation) would be beneficial.

## Figures and Tables

**Figure 1 animals-13-01382-f001:**
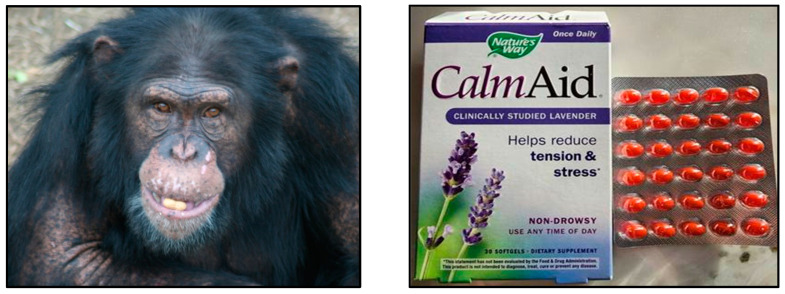
Male chimpanzee (*Pan troglodytes*), TO, who received oral lavender capsules and the CalmAid^®^, softgel capsules.

**Figure 2 animals-13-01382-f002:**
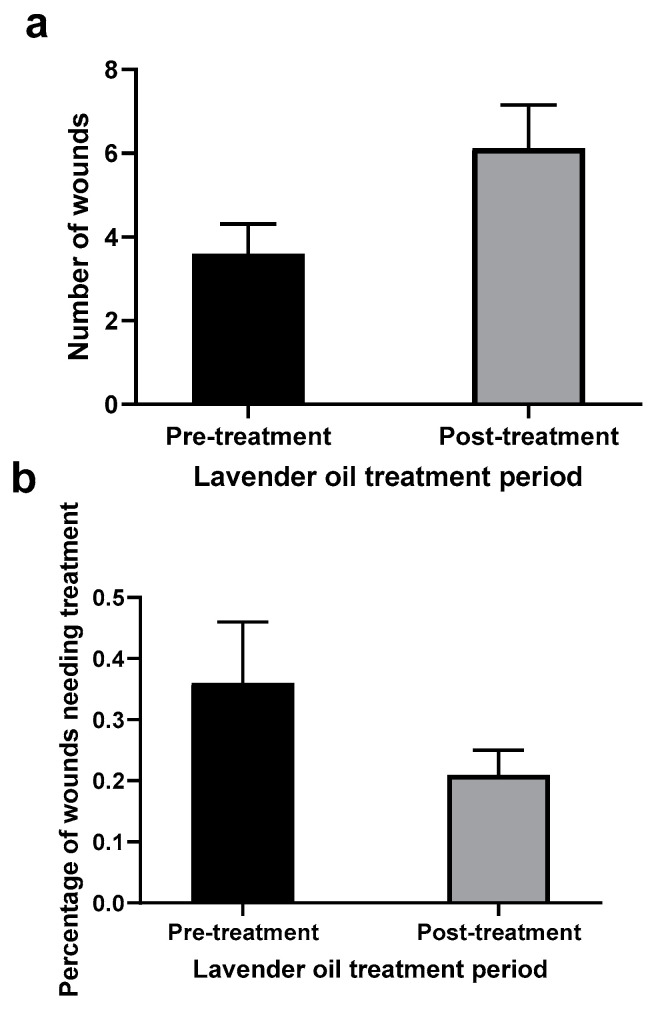
The number of overall wounds increased during lavender treatment compared to pre-treatment (*p* = 0.01). (**a**) While overall wounding increased post-treatment, (**b**) the percentage of wounds requiring treatment decreased following initiation of lavender therapy (*p* = 0.02).

**Table 1 animals-13-01382-t001:** Chimpanzee demographic information.

Initials	Sex	Age (in Years)	Rearing	Number of Chimpanzees in Group	Number of Treatment Days
**SI**	M	48	Unknown	6	476 *
**TU**	F	39	Mother	6	626
PE	F	35	Mother	6	626
KA	F	38	Nursery	6	626
CH	F	37	Nursery	6	626
NA	F	37	Nursery	6	626
**NI**	M	30	Mother	5	613
CHI	M	30	Mother	5	613
MO	F	27	Mother	5	613
BR	F	23	Mother	5	613
MA	F	34	Mother	5	613
**AJ** **	M	39	Unknown	4 *	67 *
AD	M	44	Mother	4 *	67 *
AL	F	32	Nursery	4 *	67 *
LA	F	41	Unknown	4 *	67 *
**AK**	M	38	Nursery	6	642
MAR	F	53	Unknown	6	642
MY ***	F	53	Unknown	6	n/a
ZO	F	17	Mother	6	642
TAS	F	26	Mother	6	642
CE	F	28	Mother	6	642
**TO**	M	25	Mother	5	245
BA	M	37	Mother	5	245
KE	M	37	Mother	5	245
CA	F	40	Nursery	5	245
TA	F	50	Unknown	5	245

Note: Chimpanzees receiving lavender treatment are indicated in **bold**. * SI received fewer treatment days (476) compared to the rest of his group (626), as his lavender therapy was started later than TU’s. ** AJ’s group underwent social introductions during the treatment period. Therefore, only the days on which all group members were present were counted as treatment days. *** Excluded from analyses due to death during treatment period.

**Table 2 animals-13-01382-t002:** Descriptive statistics for each wounding variable (per animal) and significance level for each pre-treatment vs. treatment period comparison.

	Mean Pre-Treatment	SEM	Mean Treatment	SEM	*p*-Value
Total number of wounds	**3.60**	**0.71**	**6.12**	**1.03**	**0.01**
Number of wounds needing any treatment	0.96	0.25	1.60	0.24	0.91
Percentage of wounds needing any treatment	**0.36**	**0.10**	**0.21**	**0.04**	**0.02**
Number of ongoing days	7.75	1.45	6.87	0.55	0.16
Number of severe wounds	3.60	0.88	5.67	0.90	0.12
Number of wounds needing antibiotic treatment	0.80	0.34	1.07	0.27	0.70
Percentage of wounds needing antibiotic treatment	0.26	0.10	0.22	0.07	0.49

Note: SEM means standard error of the mean. See method for descriptions of variables. *n* = 25 for all analyses. Significant findings are in **bold**.

## Data Availability

The data presented in this study are available on request from the corresponding author. The data are not publicly available due to protected health information.

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
