# Peer review of "The Effects of Oral Lavender Therapy on Wounding in Chimpanzees (Pan troglodytes)"

_animals, 2023, doi:10.3390/ani13081382_

Round 1

Reviewer 1 Report

In this manuscript the effect of lavender oil on the number and severity of wounds in groups of captive chimpanzees was assessed. Although of potential interest, I have major concerns about the set-up and the conclusions of this study. The fact that the number of wounds increased during the lavender therapy was not explained. Although the authors do not believe that this was induced by the lavender therapy given the abundant evidence of other results in the literature, this was a significant finding. The authors do not consider the social composition and group life of chimpanzees, which differs from that of humans and rodents. By providing lavender oil only to the instigators, the normal controls and subtle relations between the different group members might be affected, resulting in increased, albeit less violent, number of conflicts. Although the percentage of severe wounds might have decreased, the absolute number of severe wounds and the wounds needing antibiotic treatment did not differ before and after treatment. This is not addressed by the authors. Over-all, the data point more towards no effect of lavender treatment than to an overall positive effect on wounding in chimpanzees.

Reviewer 2 Report

In this manuscript, the authors describe the administration of oral lavender therapy to six chimpanzees, chosen for their aggressive tendencies. During the period in which these individuals received lavender, the number and severity of wounds (ie, requiring treatment) for all members of the social groups (n=19) was recorded. These data were then paired with a pre-administration period, matched for duration. The authors hypothesized that chimpanzees in groups receiving lavender would see a reduction in wounding. However, the number of wounds measured during the treatment period was significantly higher than in the pre-treatment; on the other hand, the percentage of wounds requiring veterinary attention was significantly lower. The content is presented clearly and succinctly. As chimpanzee wounding can be severe, any information regarding treatments without habituation/dependence/withdrawal or other side effects will be helpful for the management of chimpanzees.

My only line-specific comment is to italicize "ad libitum" at line 137. 

Regarding the study design, it seems the original focus was the administration of lavender and close examination of wounding, and that the "pre-treatment" period was determined after the fact, based on the duration of the lavender. This leads a reader to wonder whether staff were more attentive to wounds, when it was the focus of the study (before the post-hoc comparison was added to the study design); that is, whether there was consistency in attention to/recording of wounding between both time periods? A quick sentence about standardized recordkeeping should alleviate any doubt.

Was the increase in wounding (number) or decrease in wounding severity (needing treatment) noticeable before statistics were run, or were statistics run at specified intervals to determine efficacy and whether to continue use of treatment? If so, can the authors speak to how they weigh these costs and benefits (number vs severity)? Some individuals received treatment for nearly two years so a sustained increase in wounding to groupmates for that duration, despite lower severity, necessitates some justification for continued use of lavender treatment.

The authors cite literature indicating an accumulative effect of treatment -- can they specify how long it typically takes before an effect is seen (even if data only exist for humans or rodents)? If it's on the scale of years, it's not a strong argument for use in chimpanzees especially for managing short term social aggression (eg during introductions). Along those lines of dose-dependent effects, is it worthwhile to separate the dataset into subsets (0-3 months, 4-6 months, etc) to better approximate a time series analysis? Perhaps such an analysis could identify when effects of treatment first become noticeable. 

I understand that this study is meant to be an intra-group comparison (pre- and post- treatment), but I was disappointed not to see any external controls included, for example matched periods of time in other social groups, not receiving any therapy (or receiving a placebo). If it is not possible to include these data, it should be mentioned as a caveat in the Discussion as a potential shortcoming of the study design and that as such the results should be interpreted with some caution.

I think the authors do a careful job in the conclusion suggesting avenues for future research without wholeheartedly advocating the use of lavender. With more support, this is absolutely a safe and cost-effective anxiolytic that I would pursue using in managing chimpanzee aggression, so I would like to see this published to inspire more work in this area.

Reviewer 3 Report

The authors present an interesting exploratory in study in the use of lavender oil in chimpanzees to decrease wounding. This was with the perspective that if effective, decreased anxiety and stress might decrease conflict and subsequently wounding.  The authors found that although wounding events did not decrease, in fact they slightly increased during the treatment, the severity of wounding was significantly lower.  

Studies exploring  complimentary therapies like essential oils in primates as an adjunct to strong behavioral management programs are important as they present opportunities to achieve beneficial effects superior to conventional pharmacological interventions to improve welfare. The authors make this point very clearly in the introduction by giving important background on lavender oil in humans and its applications to alleviate stress and anxiety in comparison with conventional benzodiazepines which result in sedation, dependence, tolerance, and withdrawal. They selected oral lavender for this study in a commercially available formulation that had previously shown success in the human clinical situation.  They used of a standardized oral formulation which greatly increases the methodologic rigor of this study.  

While the results are equivocal, it is not surprising since the authors describe the limitations that are present in any exploratory study with only small numbers of animals (appropriately) exposed to a treatment, with historical self as a control.  While rates of wounding in any stable cohort change over time, related to many factors that the authors should mention (change in caregivers, resources offered, ect), it would be extremely difficult to control for every one of these so it was important that they look at a combination of features related to wounding and severity is certainly a key management feature.  Considering not only the wounding, but also the medical management of wounds, is a substantial welfare burden it is arguably amongst the most important variables that these authors might consider given that the wounding rate is low (and will never be zero in social primates who exhibit normal species typical behaviors in establishing rank).   

Because of low wounding in general, I understand that the authors did not treat a comparator group, but this would have been interesting and as they consider expanding the study worth evaluating as some of their results may reflect the bias of selecting for 'instigators'.  In sheep, it was shown that effects of EOs were temperament drive, having an anxiolytic effect for calm sheep, and nervous sheep showing higher agitation and plasma cortisol compared to controls in response to a stressor (Hawken et al., 2012).

The methods are well explained and the groups studied well defined, the choice to use the well-characterized formulation of EO was important - it would be good to understand when the dose was administered and the rationale for once daily, etc since this has been used BID in humans and the authors acknowledge in the discussion that dose may play a role in effect.  

In the results the authors have presented a comparison in the pre versus post period, because the number of days are variable between groups it seems that wounding could be better presented as rate, per X patient days.

The discussion is well positioned and the authors clearly acknowledge the limitations of the exploratory study design format.  

The points made regarding the importance of  empirical examination of desired effects for cost, efficiency of staff time, and asking the animals to take medications or supplements that are not beneficial is a key point and perfect way to position future studies envisioned by this team.  Given the animals enjoyed taking the treatment (cooperation as described confirms this) with combined significant reduction of wound severity, even considering the paradox of apparently increased wounding in this small cohort it should be considered for additional evaluation as the authors suggest.  Evidently the authors are exploring complimentary medicine, in this case a supplement that has a well earned reputation of 'does no harm' and essentially has no interactions with conventional drugs, and 'complimentary' could even be further emphasized to this point... specifically is not considered by itself to be standard treatment, to level expectations of readers of this type of work.

Round 2

Reviewer 1 Report

The authors answered the questions raised and improved the manuscript accordingly. Although a small decrease in the percentage of severe wounding is observed, the total number of wounds increases for unkown reasons. Therefore, the total number of severe wounds is similar in pre- and during treatment. In the revised version this is partly addressed by the authors. Although there is an indication of some effect of the treatment I am still not very convinced about the benefit of the use of lavender oil in chimpanzees to prevent severe wounding.The authors state correctly that further research is needed in larger groups of animals.

Reviewer 3 Report

Authors were responsive to reviewer comments, improving the manuscript.